# Characterization of a Common *S* Haplotype *BnS-6* in the Self-Incompatibility of *Brassica napus*

**DOI:** 10.3390/plants10102186

**Published:** 2021-10-15

**Authors:** Zhiquan Liu, Bing Li, Yong Yang, Changbin Gao, Bin Yi, Jing Wen, Jinxiong Shen, Jinxing Tu, Tingdong Fu, Cheng Dai, Chaozhi Ma

**Affiliations:** 1National Key Laboratory of Crop Genetic Improvement, College of Plant Science and Technology, National Sub-Center of Rapeseed Improvement in Wuhan, Huazhong Agricultural University, Wuhan 430070, China; lzq0826@163.com (Z.L.); blihzau@163.com (B.L.); yibin@mail.hzau.edu.cn (B.Y.); wenjing@mail.hzau.edu.cn (J.W.); jxshen@mail.hzau.edu.cn (J.S.); tujx@mail.hzau.edu.cn (J.T.); futing@mail.hzau.edu.cn (T.F.); 2College of Agriculture and Biology, Zhongkai University of Agriculture and Engineering, Guangzhou 510225, China; yangyong@zhku.edu.cn; 3Wuhan Vegetable Research Institute, Wuhan Academy of Agricultural Science, Wuhan 430345, China; gaocb1983@163.com

**Keywords:** self-incompatibility, *S* haplotype, *Brassica napus*, SCAR (sequence characterized amplified regions) marker, *SCR* (*S*-locus cysteine rich)

## Abstract

Self-incompatibility (SI) is a pollen-stigma recognition system controlled by a single and highly polymorphic genetic locus known as the *S*-locus. The *S*-locus exists in all *Brassica napus* (*B. napus*, AACC), but natural *B. napus* accessions are self-compatible. About 100 and 50 *S* haplotypes exist in *Brassica rapa* (AA) and *Brassica oleracea* (CC), respectively. However, *S* haplotypes have not been detected in *B. napus* populations. In this study, we detected the *S* haplotype distribution in *B. napus* and ascertained the function of a common *S* haplotype *BnS-6* through genetic transformation. *BnS-1/BnS-6* and *BnS-7/BnS-6* were the main *S* haplotypes in 523 *B. napus* cultivars and inbred lines. The expression of *SRK* in different *S* haplotypes was normal (the expression of *SCR* in the A subgenome affected the SI phenotype) while the expression of *Bn**SCR-6* in the C subgenome had no correlation with the SI phenotype in *B. napus*. The BnSCR-6 protein in *BnSCR-6* overexpressed lines was functional, but the self-compatibility of overexpressed lines did not change. The low expression of *BnSCR-6* could be a reason for the inactivation of *BnS-6* in the SI response of *B. napus.* This study lays a foundation for research on the self-compatibility mechanism and the SI-related breeding in *B. napus*.

## 1. Introduction

Self-incompatibility (SI), which is a genetic mechanism that helps to avoid inbreeding depression and promotes outcrossing by rejecting self-pollination, occurs in approximately 40% of flowering plant species [1,2]. The single polymorphic genetic *S*-locus regulates SI responses in many plant species [3]. In Brassicaceae, the *S*-locus mainly includes the pollen determinant of SI, *S*-locus cysteine-rich protein (*SCR*)/*S*-locus protein 11 (*SP11*) [4,5], and the stigma determinant of SI, a papilla cell localized in membrane-anchored Ser/Thr kinase (*S*-locus receptor kinase gene, *SRK*) [6,7]. Previous studies have indicated that SCR/SP11 is the ligand for SRK [8,9,10,11]. When the pollen lands on the stigma of the same *S* haplotype, specificity recognition between SCR and SRK will trigger the SI response [7,12]. This *S* haplotype-specific receptor–ligand interaction results in the activation of a pollen-inhibitory signaling pathway upon self-pollination [13]. There are approximately 100 known *S* haplotypes in *B. rapa* and 50 *S* haplotypes in *B. oleracea* that regulate SI response, respectively [14,15]. Based on the dominance and nucleotide sequences of *SCR* and *SRK* in *Brassica*, *S* haplotypes can be divided into two classes: Class Ⅰ and Ⅱ [16,17]. Genetically, *SCR* genes of class II are dominated by class I, while the *SRK* genes of these two classes are codominant [18]. Further studies have indicated that the dominant effect of different *SCR* genes is regulated by *SMI* (*SCR-methylation-inducer*) and *SMI2* (*SCR-methylation-inducer 2*) [19,20].

Most *B. napus* accessions are self-compatible, while the two progenitors, *B. rapa* and *B. oleracea*, are self-incompatible naturally [17,21]. It is generally believed that *B. napus* lost SI during its speciation by crossing *B. rapa* and *B. oleracea* [22]. Eight *S* haplotypes have been identified in *B. napus*, named *BnS-1* to *BnS-7* and *BnS-1300*, among which *BnS-1* to *BnS-5* are Class I haplotypes and *BnS-6*, *BnS-7* and *BnS-1300* are Class II haplotypes [17,23]. Recent studies have shown that the *S* haplotype in the C subgenome of *B. napus* is *BnS-6*, which is the homolog of *BoS-15* in *B. oleracea* [17,24]. The main *S* haplotype in the A subgenome of *B. napus* was found to be *BnS-1.* Pollination testing and Northern blotting showed that the SC (self-compatibility) of *B. napus* ‘Westar’ with *BnS-1/BnS-6* was caused by the non-expression of *BnSCR-1* [17]. Further studies have shown that a 3.6 kb *Helitron*-like transposon insertion in the promoter region of *BnSCR-1* leads to the inactivation of *BnSCR-1* in ‘Westar’ [25,26]. This *Helitron*-like transposon was found to be widely present in *B. napus* with *BnS-1* and was transposed to the current location during the speciation of *B. napus* [26]. Moreover, the loss of function of *SCR* in pollen and translational repression might cause the SC of *B. napus* with *BnS-7/BnS-6* [24]. However, the function of *BnS-6* in the C subgenome has not been well studied.

In *B. napus*, SI is one of the main approaches for heterosis utilization. The *B. napus* line ‘S-1300’, a synthetically developed double-low *B. napus* self-incompatible accession [27], contains two class II *S* haplotypes: *BnS-1300* and *BnS-6* [23,28]. The SI of ‘S-1300’ is recessive to *BnS-1/BnS-6* lines but dominant to *BnS-7/BnS-6* lines [23,24,29]. A perfect tribe-cross hybrid seed produce system was established by combining ‘S-1300’ with *BnS-1/BnS-6* lines (SI restorer lines) and *BnS-7/BnS-6* lines (SI maintainer lines) [24,30]. However, the self-incompatible accessions in floral morphologies are similar to the self-compatible accessions, meaning that it is difficult to discriminate the contaminated plants in SI breeding lines [23]. Thus, the development of *S*-locus-linked genome markers would allow practical detection across the whole growth period. Here, we developed stable SCAR markers based on *BnS-1300*, *BnS-1*, *BnS-6*, and *BnS-7* and confirmed that the major *S* haplotypes in 523 *B. napus* cultivars and inbred lines were *BnS-1/BnS-6* and *BnS-7/BnS-6*. RNA expression analysis showed that the *SRK* of all *S* haplotypes was codominant and that the expression of *SCR* determined the SI phenotype in *B. napus*. The further functional characterization of *BnS-6* showed that *BnSCR-6* is functional, but in *B. napus* and *B. oleracea* with *BoS-15*, there may be abnormalities in the SI signal pathway or recognition specificity. These results provide stable molecular markers for the SI-related breeding of *B. napus* and insight into the molecular basis of loss of SI in *B. napus.*

## 2. Results

### 2.1. BnS-1/BnS-6 and BnS-7/BnS-6 Are the Main S Haplotypes in B. napus

The sequence collinear comparison of *SCR* (Appendix A) and *SRK* of the *S* haplotypes of ‘S-1300’, its restorer line ‘Westar’ and its maintainer line ‘Bing409’, indicated that the *S* haplotypes in the A subgenome of ‘S-1300’, ‘Westar’, and ‘Bing409’ were *BnS-1300*, *BnS-1*, and *BnS-7*, respectively, while the *S* haplotypes in the C subgenome of these three lines were *BnS-6*. Therefore, we developed six pairs of SCAR markers in order to identify the *SCR* and *SRK* genes of *BnS-1300*, *BnS-1*, and *BnS-7* in the A subgenome. The corresponding PCR products of different *SCR* and *SRK* genes were found to be 473 bp (*BnSCR-1300*, Gene accession number XR_004449204) and 472 bp (*BnSRK-1300*, Gene accession number AB097116), 412 bp (*BnSCR-1*, Gene accession number AB270773) and 883 bp (*BnSRK-1*, Gene accession number AB086976), and 353 bp (*BnSCR-7*, Gene accession number AB270770) and 294 bp (*BnSRK-7*, Gene accession number AB008191), respectively (Figure 1).

To verify the effect of the SI-related SCAR molecular markers, the ‘S-1300’ line was crossed as the female parent with 14 self-compatible inbred lines to produce 14 F_1_ generation hybrids. First, the marker SRK6-1 [23] was verified in these self-compatible lines. As shown in Table 1 and Appendix A, all these materials contained the SRK6-1 products, indicating that the *S* haplotypes in the C subgenome of all these lines were *BnS-6* or *BnS-6*-like. Then, the markers of *BnS-1300*, *BnS-1*, and *BnS-7* were tested in these 14 lines. In the A subgenome of ‘Westar’, ‘Huangshuang2’, ‘Tapidor’, and ‘89008’, the *S* haplotype was *BnS-1*, while in the A subgenome of ‘Ningyou-7’, ‘326’, ‘614’, ‘1728’, ‘C32’, ‘Bing409’, ‘242’, ‘198’, ‘1745’, and ‘230’, the *S* haplotype was *BnS-7* (Table 1). To further confirm whether the genotype was associated with the SI phenotype, the F_1_ plants were self-pollinated and the SC Index (SCI) of F_1_ plants of ‘Westar’, ‘Huangshuang2’, ‘Tapidor’, and ‘89008’ were crossed with ‘S-1300’ ranging from 17.68 to 21.06, which were presented as the SC phenotype. The SCI of F_1_ plants of the other ten lines crossed with ‘S-1300’ ranged from 0.02 to 0.64, which were presented as the SI phenotype (Table 1). These results indicated that these SCAR markers were able to identify the *S* haplotype in *B. napus*.

The *S*-locus was found to widely exist in *B. napus* lines, however, the distribution of *S* haplotypes in *B. napus* natural population was still unclear. We then investigated the *S*-locus in 523 *B. napus* inbred lines and cultivars using the present SCAR markers. In 523 *B. napus* inbred lines, 239 lines contained *BnS-1* (45.70%); 226 lines contained *BnS-7* (43.21%); and only 3 lines (‘11-P63-5Y7’, ‘11-P63-8Y32’, and ‘11-P63-3Y3’) contained *BnS-1300*. Interestingly, 20 lines contained the *BnS-1* and *BnS-7* haplotypes simultaneously, suggesting that more than one *S*-locus could exist in the A subgenome of these *B. napus* lines. There were also 35 cultivars that could not be detected in any mentioned *S* haplotypes, indicating that there were other *S* haplotypes in these 35 cultivars. We also tested the *S* haplotype in the C subgenome and the results indicated that all 523 lines were *BnS-6* (Table 2 and Appendix A). Obviously, these 239 *BnS-1/BnS-6* cultivars and 226 *BnS-7/BnS-6* cultivars could be used as restorer and maintainer lines for ‘S-1300’ in SI-related breeding, respectively.

### 2.2. Gene Expression of SRK and SCR in Different S Haplotypes

*SRK* and *SCR* are the key components of SI response, so we designed specific qRT-PCR primers (Appendix A) based on the *SRK* and *SCR* of different *S* haplotypes in order to research the SC mechanism of *B. napus*. As only 5 bp differences are included between the CDS of *BnSCR-1300* and *BnSCR-6* (Genebank accession number AB270774), it is difficult to distinguish them as one bp artificial mismatch was introduced at the 3′ end of the reverse primer apart from the internal 2 bp substitutions (Appendix A). Using these primers, in addition to the specific primers of *BnSCR-1* [23] and *BnSCR-7* [24], we detected the RNA expression of *BnSCR-1*, *BnSCR-1300*, *BnSCR-6*, and *BnSCR-7* in different tissues of corresponding materials. *SCR* genes were specifically expressed in the anther of self-incompatible lines and the expression level increased with the development of the anther. In ‘S-1300′, the expression level of *BnSCR-1300* was consistent with that of ‘BrHB’, a *BrS-60* contained *B. rapa* SI line (Appendix A), and reached the highest relative expression level of 6.12 in the L4 anther (Figure 2a). In the transgenic SI line ‘W-3’, the expression pattern of *BnSCR-1* (Figure 2c) was similar to that of *BnSCR-1300* in ‘S-1300’. However, in the SC lines ‘Westar’ and ‘Bing409’, the transcript levels of *BnSCR-1* and *BnSCR-7* were near 0 (Figure 2b,c). The expression levels of *BoSCR-15* in the L4 anther of *B. oleracea* was 0.407 (Appendix A). *BnSCR-6* is a homolog of *BoSCR-15* and exists in all *B. napus* lines, but the expression levels of *BnSCR-6* in all *B. napus* samples were very low (Figure 2a–c). Additionally, the expression levels of *BnSRK-1*, *BnSRK-1300*, *BnSRK-6* (Genebank accession number AB270772), and *BnSRK-7* in the mature stigma of all *B. napus* lines mentioned above were detected. In the stigma of all *B. napus*, the expression levels of *SRK* were increased along with the development of buds, and reached the highest level in the L4 stigma, but were not associated with SI. The relative expression levels of *SRK* in the L4 stigma ranged from 0.211 to 0.661 (Figure 2d–f).

The results revealed that the expression patterns of *SCR* and *SRK* from different *S* haplotypes were similar in *B. napus, B. rapa*, and *B. oleracea*. They were not expressed in vegetative organs but were highly expressed in reproductive organs. With the development of flowers, the expression levels of *SCR* and *SRK* gradually increased in the anthers and stigma, respectively. The expression levels of all *SRK* were normal and had no corresponding relationship with the SI phenotype. However, the expression levels of *SCR* in SI lines (*BnSCR-1300* in ‘S-1300’ and *BnSCR-1* in ‘W-3’) were three orders of magnitude higher than those in SC lines (*BnSCR-1* in ‘Westar’ and the expression level of *BnSCR-7* in ‘Bing409’). The above results indicate that the low expression of *SCR* is the reason for the SC in ‘Westar’ and ‘Bing409’.

### 2.3. Functional Validation of BnSCR-6

*BnS-6* existed in all 523 *B. napus* lines, and *SRK* was normally expressed. In order to investigate the function of *BnS-6* in the SI response of *B. napus*, a 1975 bp fragment that included the promoter and full coding region of *BnSCR-6* was obtained from ‘S-1300’ and 14 self-compatible *B. napus* lines in the SCAR marker development section. The sequences of *BnSCR-6* in these lines and *BnSCR-15* in *B. oleracea* [31] were conserved. Motif search using PlantCARE (http://bioinformatics.psb.ugent.be/webtools/plantcare/html/, accessed on 18 September 2015) revealed that there were no obvious differences in the predicted *cis*-elements in the promoters of *BnSCR-6* and *BnSCR-1300* (Appendix A). Then, the 1398 bp promoter of *BnSCR-6* was fused to the GUS reporter construct and introduced into wild-type *Arabidopsis thaliana* (Col-0) plants. The GUS staining results showed that the promoter was functional and specific in the middle buds and was similar to the promoter of *BnSCR-1300* (Figure 3). A series of promoter deletion cassettes of *BnSCR-6* and *BnSCR-1300* were also analyzed, but GUS expression could still be detected in the middle buds (Figure 3). These GUS staining results confirm the functional consistency of the promoters of *BnSCR-6* and *BnSCR-1300.*

As described above, *BnSRK-6* was normally expressed. To rebuild the SI response of *BnS-6* in *B. napus*, the CDS of *BnSCR-6* was expressed in ‘Westar’ under the control of 1851 bp and 684 bp promoters of *BrSCR-47*, a Class I dominant SCR in *B. rapa* s (Figure 4a) [26]. At least 20 independent transgenic lines were generated in the 1851 bp and 684 bp promoter constructs, respectively. Both the T_1_ and T_2_ progenies were self-compatible according to the pollination assay (Figure 4c–e). The SC phenotype was also observed in seven *B. oleracea* with *BoS-15* (Appendix A). The phenotypes were compatible when pollinated with the pollen of *BnSCR-6* overexpressed lines on the stigma of *B. napus* with *BnS-6* and that of *B. oleracea* with *BoS-15* (Figure 3c–f, Appendix A). These results indicated that the loss of SI of *BnS-6* (*BoS-15*) was a common phenomenon in *B. napus* and *B. oleracea* lines, and that only normally expressed *BnSCR-6* could not recover the SI response.

We hypothesize that the recognition between *BnSCR-6* and *BnSRK-6* or the downstream signal pathway may cause an abnormal SC phenotype of *BnS-6*. Pollination assays were performed between the T_2_ or T_3_ progeny of *BnSCR-6*-overexpressed lines with other *B. napus* lines. Interestingly, when the pollen of *BnSCR-6*-overexpressed lines in T_3_ progeny were pollinated to the stigma of *B. napus* with *BnS-6*, little-seed-setting appeared in the pollination between nine of fifteen individuals with ‘S-1300’, and all the seed-setting was normal in the pollination between the fifteen individuals with ‘Ningyou-7’, ‘326’, ‘Bing409’, and ‘ZS11’ (Figure 5). These results confirmed that BnSRK-1300 in ‘S-1300’ could recognize BnSCR-6 in *BnSCR-6*-overexpressed lines, which was consistent with the results of previous studies on *BrSRK-60* and *BoSCR-15* in *B. rapa* and *B. oleracea* [32,33]. The above results indicated that *BnSCR-6* was functional in *BnSCR-6*-overexpressed lines, but there could be abnormalities in terms of the SI signal pathway or the recognition specificity in *B. napus* and *B. oleracea* with *BoS-15*.

## 3. Discussion

SI is an important biology phenomenon. The SI response in *Brassica* is regulated by SCR/SRK recognition. However, the distribution of the *S* haplotype in *B. napus* population is not very clear, which has impeded the research on the SC of *B. napus*. Through the development of SI-related SCAR markers, we analyzed a *B. napus* natural population and found that *BnS-1/BnS-6* and *BnS-7/BnS-6* were the most common *S* haplotypes in the *B. napus* population. Additionally, we preliminarily analyzed the function of *BnSCR-6*, finding that the low expression of *BnSCR-6* is one of the reasons for the inactivation of *BnS-6* in the SI of *B. napus*. The above results lay a foundation for the further analysis of the SC mechanism in *B. napus*.

SI is a mechanism in plants that prevents inbreeding through the rejection of self-pollen [34]. *B. napus* is an important oil crop around the world and heterosis is the main way to increase its yield and quality [35]. Compared to the cytoplasmic male sterility system, there are some advantages to the self-incompatible system, such as shorter breeding periods required, a wider range of restorer lines, and no negative cytoplasmic effects [23,36], which is a benefit for *B. napus* breeding. However, due to the normal flower phenotype, it is hard to distinguish the self-incompatible and self-compatible lines. The design of molecular markers based on the *S*-locus is very important for the SI-related breeding of *B. napus*. In the past decade, several draft molecular markers have been developed [23,28,37]. There are also some issues with the existing molecular markers, such as the results not being reproducible and the need for several PCR programs to be designed for different markers, which is time- and labor-consuming. Based on the *S* haplotypes in the A subgenome, we designed SCAR markers specifically for the amplification of the *SCR* and *SRK*. These SCAR markers could help to identify the *S* haplotype of *B. napus* in a precise and efficient manner. In 523 *B. napus* cultivars and inbred lines, we identified many *B napus* lines that could be used as restorer lines (239 individuals, *BnS-1/BnS-6*) or maintainer lines (226 individuals, *BnS-7/BnS-6*) of ‘S-1300’ in the breeding process. In the breeding process, the maintainer lines and restorer lines found in this study could improve the SI line ‘S-1300’ and obtain lots of hybrid combinations with potential agricultural traits. In addition, these SCAR markers could accelerate the breeding process and commercial SI hybrid seeds production.

There are about 100 *S* haplotypes in *B. rapa* and 50 *S* haplotypes in *B. oleracea* [14,15]. However, only eight *S* genotypes have been reported in *B. napus* [17,23]. Recent studies have shown that the major *S* haplotypes are *BnS-1/BnS-6* and *BnS-7/BnS-6* and that the *S* haplotype in the C subgenome of all *B. napus* is *BnS-6*. This is consistent with previous reports that there are six *S* haplotypes in the *B. napus* A subgenome, most of which are *BnS-1* [17]. Three cultivars originating from ‘S-1300’ are *BnS-1300/BnS-6* (Table 2 and Appendix A). Twenty cultivars were found to have both *BnS-1* and *BnS-7* in the A subgenome (Table 2 and Appendix A), suggesting more than two *S* haplotypes could exist in these *B. napus* lines. However, there are also 35 cultivars that couldn’t detect any *S* haplotype could be detected in the A subgenome, indicating that there were other *S* haplotypes. *B. napus* (AACC), a young allotetraploid species, was derived from the hybridization of two diploid species, *B. rapa* (AA) and *B. oleracea* (CC), about 7500 years ago [38]. However, only a few allotetraploidization events have occurred and been stably passed on to *B. napus* [39], which may have resulted in fewer *S* haplotypes existing in *B. napus.*

*B. napus* is self-compatible, although *S*-locus is present. The SC of *B. napus* with *BnS-1/BnS-6* is due to the presence of a transposon insertion in the *BnSCR-1* promoter position in the A subgenome, which results in the abnormal expression of *BnSCR-1* [17,25,26]. The *S* haplotype in the C subgenome of all-natural *B. napus* accessions is *BnS-6*, and the *SCR* and *SRK* sequences are consistent with the homologous *S* haplotype *BoS-15* in *B. oleracea*. The results of the RNA expression analysis show that all *SRKs* were normal, but *BnSCR-6* was expressed in barely any tissues of *B. napus*. The sequences of *SCR* and *SRK* of *BnS-7* in *B. napus* are consistent with the homologous *gene sequences* in *B. rapa*, and *BnSRK-7* is normally expressed, but all these lines are self-compatible. Although *BnSCR-7* of some materials is expressed [24], the level is far lower than that of *BnSCR-1300* in ‘S-1300’ and *BnSCR-1* in ‘W-3’ (Figure 2). Such differences in expression levels are likely to be the cause of phenotypic differences. Therefore, the difference in *SCR* expression is one of the direct causes of the self-compatible phenotype of *B. napus*.

The further comparison of *BnSCR-6* and *BnSCR-1300* showed that the predicted *cis*-element and GUS activation of *BnSCR-6* and *BnSCR-1300* were similar; meanwhile, the *BnSCR-6*-overexpressed lines were self-compatible in multigenerational self-pollination. Hadj-Arab et al. [40] found self-compatible individuals in a multigenerational self-pollinated *B. oleracea* with *BoS-15*, and additional self-compatible individuals emerged from the self-progenies. RNA expression analysis revealed that *BoSCR-15* was not expressed in these self-compatible individuals, which accounts for the SC of these individuals [40]. In addition, *B. oleracea* is not strictly self-incompatible [41,42,43,44]. Combination with the present result for the *BnSCR-6*-overexpressed lines indicated that the inactivation of *BoS-15* in the C subgenome may have occurred in the evolution of the *B. oleracea* C genome itself, rather than in the progress of *B. napus* formation.

The BnSCR-6 protein in the *BnSCR-6* overexpressed lines can recognize BnSRK-1300, resulting in partial incompatibility in the cross-pollination test (Figure 5). This shows that BnSCR-6 and BnSCR-1300 have similar recognition specificity and that differences in their sequences do not change their recognition specificity. This result is consistent with the pollination experiment in *B. rapa* and *B. oleracea* carried out by Sato et al. [33], which indicates that the function of *BnSCR-6* in the overexpressed lines is normal and that the reason why SCR and SRK of *BnS-6* cannot recognize each other in natural *B. napus* is not only due to the abnormal *BnSCR-6* expression. There are multiple reasons for the SC of *BnS-6* in *B. napus*.

In conclusion, we developed stable SCAR markers for *S* haplotype identification in *B. napus* and subsequently found that all the *S* haplotypes in the C subgenome are *BnS-6*, while *BnS-1/BnS-6* and *BnS-7/BnS-6* are the main *S* haplotypes in *B. napus*. The expression levels of *SCR* and *SRK* in different *S* haplotypes were deduced from qRT-PCR detection, which indicated that the expression level of *SRK* did not affect the SI phenotype. The expression of *SCR* in the A subgenome affected the SI phenotype, and the expression of *BnSCR-6* in the C subgenome of all materials was low. The further promoter and functional analysis of *BnSCR-6* revealed that the low expression of *BnSCR-6* is one of the main reasons for the inactivation of *BnS-6* in the SI of *B. napus*. The present work provides stable molecular markers for SI-related breeding in *B. napus* and lays the foundation for the research of the SC mechanism in *B. napus*.

## 4. Materials and Methods

### 4.1. Plant Materials and Growth Conditions

The *B. napus* SI line ‘S-1300’, its restorer line ‘Westar’, and its maintainer line ‘Bing409’ were crossed with each other to produce F_1_ progeny. ‘S-1300’ was also used as a female parent and crossed with another 12 *B. napus* SC inbred lines (Table 1) to produce F_1_ progeny for SI-related SCAR marker development. A rapeseed natural population, consisting of 523 inbred lines and cultivars from 10 countries (Appendix A) [45], was used to survey the *S* haplotype distribution in *B. napus*. The wild-type *B. napus* line ‘Westar’ was used for a test of the overexpression of *BnSCR-6*. The transgenic self-incompatible *B. napus* line ‘W-3’ was used for *BnSCR-1* expression detection. *B. napus* plants were grown in the field at Huazhong Agricultural University. *Arabidopsis thaliana* plants (Col-0) were grown at 22 °C with a 16/8 h light/dark cycle in a greenhouse.

### 4.2. SI Phenotype Assay and Pollination Assay

SI phenotype and SCI were determined following a previously described method [28]. When 3–5 flowers were present on the major inflorescence, the open flowers were removed, the major inflorescence and 2–3 secondary ramifications were covered by paper bags and kept for two weeks. After removing the bags, the seeds and flowers were counted, and the SCI was calculated as the number of seeds divided by the number of flowers. Approximately 100–150 flowers from each plant were investigated. SI phenotype of each plant was categorized as SCI < 2 (self-incompatible), and SCI ≥ 2 (self-compatible).

Pollination assay was preformed following a previously described method [46]. Floral buds of the *B. napus* were emasculated one day before anthesis to avoid pollen contamination. Pollination was performed on the anthesis day. Some pollinated pistils were left to set seeds. The rest were cut at the peduncle 16 hours after pollination, fixed for 2 h in ethanol:acetic acid (3:1), softened in 1 mol/L NaOH at 60 °C for 1.5 h and stained with 0.01% (*w*/*v*) decolorized aniline blue for 2.5 h in 2% (*w*/*v*) K_3_PO_4_. Pistils were gently squashed on a microscopic glass slide by placing the cover glass over the pistils. Samples were examined using a fluorescence microscope (Ax 10, Zeiss, Jena, Germany).

### 4.3. Sequence Collinear Comparison and Primer Design

The sequence alignment analysis of *BnSCR-1300*, *BnSCR-6*, and *BnSCR-7* was performed in Clustal Omega (https://www.ebi.ac.uk/Tools/msa/clustalo/, accessed on 26 August 2015) with the default parameters and edited with genedoc (http://nrbsc.org/gfx/genedoc, accessed on 26 August 2015). The primers were designed in the region of sequence differences using the Primer premier 6 software (http://www.premierbiosoft.com/primerdesign/index.html, accessed on 26 August 2015) with adjustment manually.

### 4.4. DNA Extraction and Genotyping Assay

Genomic DNA from all individuals was extracted from young leaves. To increase the efficiency and reduce the genotyping costs, total genomic DNA was extracted by a simple approach [47] using the following procedures: (1) Adding a stainless-steel bead and 300 µL of DNA extraction buffer (10 mM Tris–HCl, 1 mM EDTA, pH 8.0) to each well and sealing the plate with a silicone cover. (2) Grinding the leaf sample with a paint shaker for 3 min and separating the supernatants by centrifugation. (3) Transferring 100 µL of supernatants to a V-bottom plate prefilled with 50 µL of isopropanol per well, which was then mixed by pipetting, and stored at −20 °C for 30 min. (4) Precipitating the DNA by centrifuging the plate and discarding the supernatant by inverting the plate. (5) Adding 100 µL 75% ethanol and repeating step (4) and tap-drying the plate with paper. (6) Air-drying the samples overnight and resuspending the samples in 50 µL dd H_2_O. For the SCAR molecular marker assay, 10 μL reaction volume was used, which contained 5 μL 2× Taq Reaction Buffer (containing Mg^2+^, dNTPs, and DNA polymerase) (Vazyme, Nanjing, China), 0.5 μM of each primer, and 1 μL genomic DNA. The PCR reaction was performed in a T100thermocycler (Bio-Rad Laboratories, Inc., Hercules, CA, USA) using the following program: 3 min at 94 °C, 35 cycles at 94 °C for 30 s, primer annealing temperature for 30 s, 72 °C for 45 s, followed by 72 °C for 10 min. The PCR products were analyzed by electrophoresis on 1.0% agarose gel in 0.5× TAE buffer and were visualized by staining with ethidium bromide.

### 4.5. RNA Extraction and Quantitative Real-Time PCR (Qrt-PCR)

Roots, leaves, stems, flower buds (L1: 0–2 mm buds), anthers (L2–L4 anther: anther in 2–4, 4–6, and 6–8 mm buds), and stigma (L2–L4 stigma: stigmas in 2–4, 4–6, and 6–8 mm buds) were collected for RNA extraction. Total RNA was extracted using the SV Total RNA Isolation System (Promega, Madison, WI, USA). The RNA samples were quantified using a NanoDrop Spectrophotometer (Nanodrop Technologies, Wilmington, DE, USA), and the first-strand cDNA was synthesized by reverse transcription with a Thermo RT kit (Thermo Fisher, Waltham, MA, USA). qRT-PCR was performed in triplicate for each sample using the SGExcel FastSYBR MasterMix (Sangon Biotech, Shanghai, China) on a CFX96 Real-Time System (Bio-Rad Laboratories, Inc., Hercules, CA, USA). Gene-specific primers used in the amplification are listed in Appendix A, and *Actin* (GenBank accession no.: AF111812) was used as an internal control to normalize the transcript levels for all the expression analyses. Relative expression levels of *SRK* and *SCR* in different *S* haplotypes were determined using the comparative 2^−ΔCT^ method and normalized to *Actin*, then the relative expression level of *BnSCR-6* in overexpressed lines was determined using the comparative 2^−ΔΔCT^ method and normalized to *BnSCR-6* in ‘Westar’ [48].

### 4.6. Promoter Construct and GUS Assay

The promoter sequences of *BnSCR-6* and *BnSCR-1300* were amplified from the DNA of ‘Westar’ and ‘S-1300’ by PCR using specific Appendix A. For the GUS assay, the promoter sequences of *BnSCR-6* and *BnSCR-1300* of different lengths were amplified from the DNA of ‘Westar’ and ‘S-1300’ and cloned into the vector pC2300-GUS [26] to yield *SCR6*-P1-GUS to *SCR6*-P4-GUS and *SCR1300*-P1-GUS to *SCR1300*-P4-GUS constructs, respectively. All promoter-GUS constructs were introduced into *A. thaliana* by *Agrobacterium*-mediated transformation. The GUS activity was visualized by staining different tissues of the T_3_ generation homozygous transgenic lines overnight in X-Gluc solution [49], then the tissues were cleaned in 75% (*v*/*v*) ethanol and imaged under a stereomicroscope (Nikon, SMZ25, Tokyo, Japan).

### 4.7. Vector Construction of Bnscr-6 Overexpression and Plant Transformation

The 1851 bp and 684 bp *BrSCR-47* promoter sequences were obtained from our previous study [26], and subcloned into pCAMBIA2300 to yield 2300::1851P and 2300::684P constructs, respectively. The CDS of *BnSCR-6* was amplified from cDNA of ‘S-1300’ using gene-specific primers, confirmed by sanger sequencing, and subcloned into 2300::1851P and 2300::684P constructs to yield 2300::1851P-SCR6 and 2300::684P-SCR6 constructs, respectively. The 2300::1851P-*SCR6* and 2300::684P-*SCR6* constructs were introduced into *Agrobacterium tumefaciens* GV3101 host cells and transformed into *B. napus* ‘Westar’ following the previous method [50]. The DNA of transformed plants were analyzed by PCR, combining the primers 47pro-1 and SCR6-R to verify the presence of *BnSCR-6* transgene.

## Figures and Tables

**Figure 1 plants-10-02186-f001:**
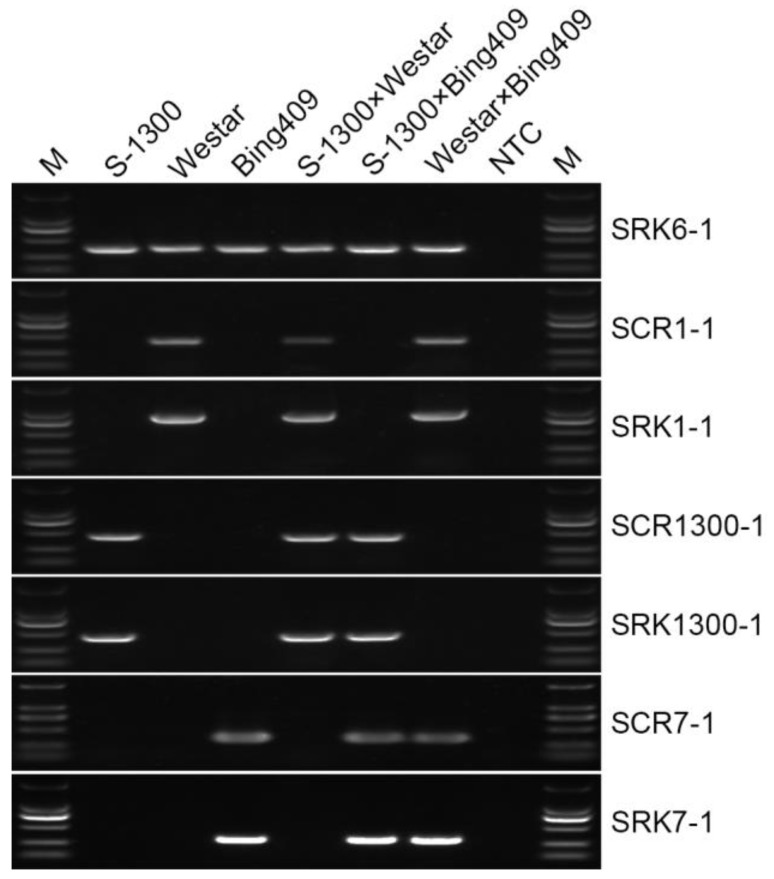
PCR fragment amplified and obtained from three kinds of parent materials and F_1_ generation of one-by-one pairs using SCAR markers. NTC: no template control. M: DNA marker, from top to bottom, the size of band was 2000, 1000, 750, 500, 300, and 200 bp, respectively.

**Figure 2 plants-10-02186-f002:**
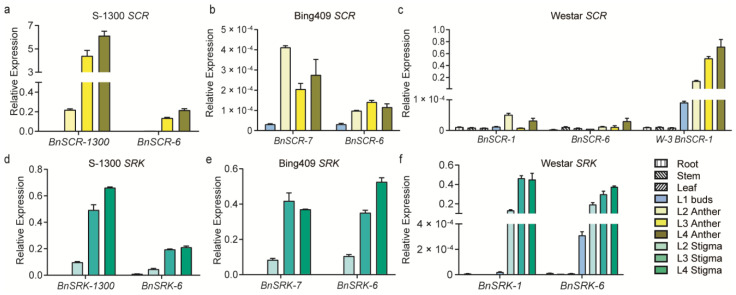
Relative expression of *SCR* and *SRK* in different tissues of *B. napus*. (**a**–**c**) Relative expression of *SCR* in different tissues of ‘S-1300’, ‘Bing409’, ‘Westar’, and ‘W-3’, respectively; (**d**–**f**) Relative expression of *SRK* in different tissues of ‘S-1300’, ‘Bing409’, and ‘Westar’, respectively. The relative expression was corrected using the reference gene *BnActin7*.

**Figure 3 plants-10-02186-f003:**
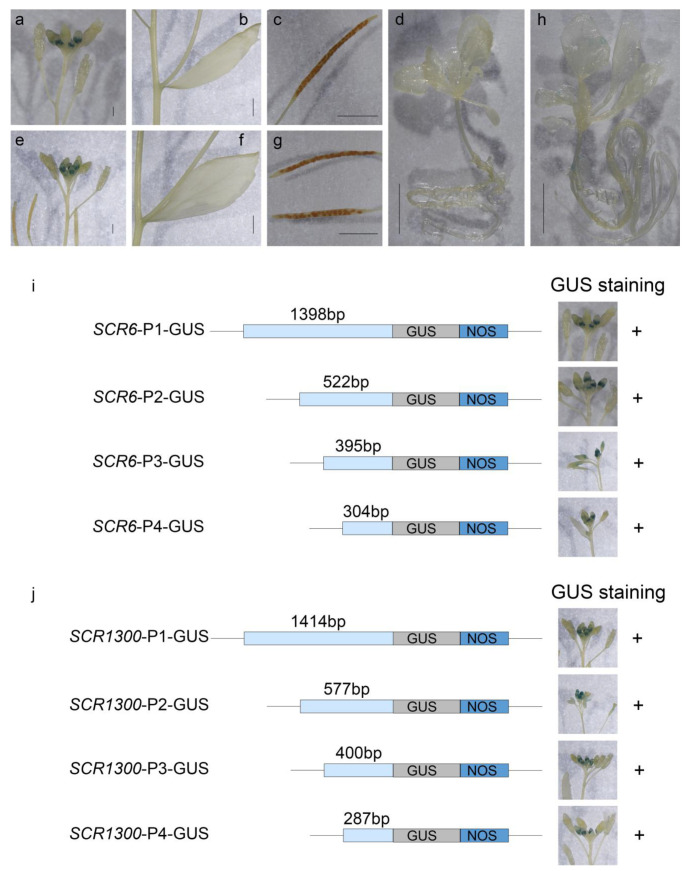
Comparison of the GUS staining of *BnSCR-6* and *BnSCR-1300* promoters in different *Arabidopsis* tissues. (**a**–**d**) are the GUS staining of *BnSCR-6* in different parts of plants; (**e**–**h**) are the GUS staining of *BnSCR-1300* in different parts of plants; (**a**,**e**) are flower buds; (**b**,**f**) are stems and leaves; (**c**,**g**) are siliques; (**d**,**h**) are the whole plant at the seedling stage; (**i**) is the GUS staining of different length of *BnSCR-6* promoter in flower buds; (**j**) is the GUS staining of different lengths of *BnSCR-1300* promoter in flower buds.

**Figure 4 plants-10-02186-f004:**
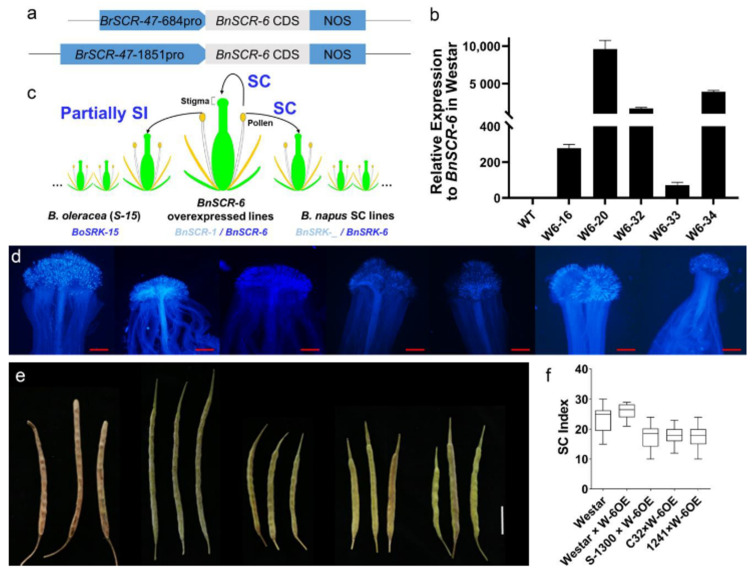
Pollination test of *BnSCR-6* overexpression lines. (**a**) Overexpression of *BnSCR-6* vector diagram. (**b**) *BnSCR-6* expression level detection in T_0_ generation. (**c**) Pollination pattern diagram. (**d**) Typical pollen tube germination in T_2_ generation, left to right: Westar, ‘Westar × We-6OE, S-1300 × We-6OE, C32 × We-6OE, 1241 × We-6OE, Gan-64 × We-6OE, Gan-118 × We-6OE. Gan-64 and Gan-118 is the *B. oleracea* material with *S-15*, Bar = 500 μm. (**e**) The corresponding siliques in Figure d, left to right: Westar, Westar × We-6OE, S-1300 × We-6OE, C32 × We-6OE, 1241 × We-6OE. Due to the poor seed-setting ability of *B. oleracea* itself, the hybrid siliques develop abnormally, Bar = 2 cm. (**f**) The SCI statistics of pollination tests in (**e**).

**Figure 5 plants-10-02186-f005:**
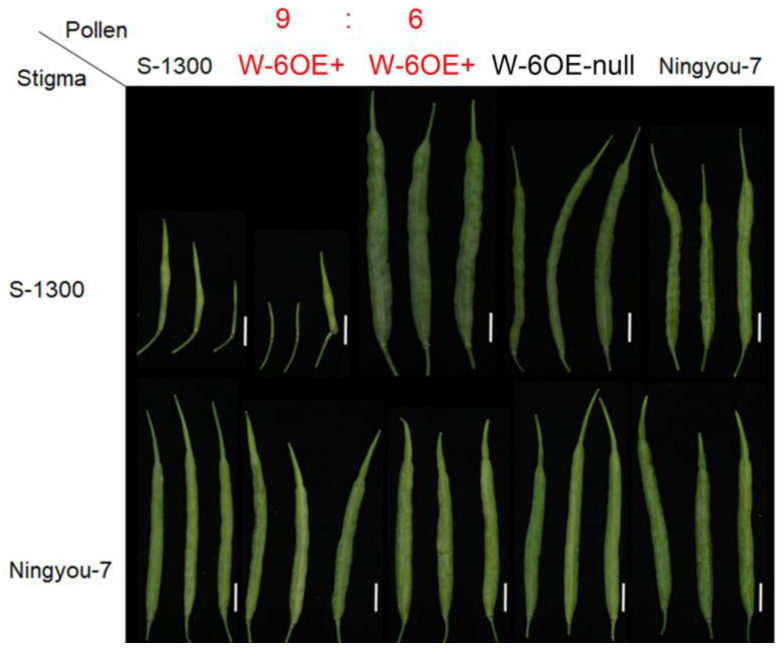
Pollination test of the T_3_ generation of *BnSCR-6* overexpression materials. W-6OE-null: genetically modified negative material; Bar = 1 cm.

**Table 1 plants-10-02186-t001:** Genotypes of the 14 *B. napus* lines collected and the phenotypes of the corresponding F_1_ hybrids.

Material	Origin	SRK1300-1/ SCR1300-1	SRK1-1/ SCR1-1	SRK7-1/ SCR7-1	SRK6-1	SI Phenotype ^1^	
						SCI	SC/SI
S-1300	China	+	−	−	+	0.01	SI
Huashuang2	China	−	+	−	+	21.06	SC
Westar	Canada	−	+	−	+	16.95	SC
Tapidor	Europe	−	+	−	+	20.60	SC
89008	China	−	+	−	+	17.68	SC
Ningyou-7	China	−	−	+	+	0.02	SI
326	China	−	−	+	+	0.31	SI
614	China	−	−	+	+	0.50	SI
1728	China	−	−	+	+	0.15	SI
C32	China	−	−	+	+	0.23	SI
Bing409	China	−	−	+	+	0.41	SI
242	China	−	−	+	+	0.64	SI
198	China	−	−	+	+	0.34	SI
1745	China	−	−	+	+	0.21	SI
230	China	−	−	+	+	0.05	SI

^1^: SI phenotypes of F_1_ generation plants from crossing ‘S-1300’ with male parents; +: successful amplification; −: no amplification.

**Table 2 plants-10-02186-t002:** Genotype statistics of 523 *B. napus* cultivars and inbred lines.

Genotypes of the Lines				Number of Lines	Proportion
SRK1300-1	SRK1-1	SRK7-1	SRK6-1		
+	−	−	+	3	0.57%
−	+	−	+	239	45.70%
−	−	+	+	226	43.21%
−	+	+	+	20	3.82%
−	−	−	+	35	6.69%

+: successful amplification; −: no amplification.

## Data Availability

Not applicable.

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
