# Peer review of "Characterization of a Common S Haplotype BnS-6 in the Self-Incompatibility of Brassica napus"

_plants, 2021, doi:10.3390/plants10102186_

Round 1
Reviewer 1 Report
This manuscript sheds additional light on the phenomenon of self-incompatibility Vs self-compatibility in various species of Brassica. I believe the work to be relevant and interesting, especially as hybrid production schemes are developed that can overcome the drawbacks of CMS. The authors made a significant effort to identify markers for the SCR and SRK components of self-compatibility in B. napus. These markers indicated that even though the factors that should result in self-incompatibility in B. napus are present, natural accessions are self-compatible. Although one of the factors was shown to be naturally down-regulated in its expression, it was shown that even when this gene was over-expressed transgenically, the resulting plants were still self-compatible. This would indicate additional factors must have evolved in the original hybridization between B. rapa and B. oleracea to result in B. napus being self-compatible, as both of these species are self-incompatible. This work sta rts to shed some light on this phenomenon by eliminating the simplest explanation, that being that one of the genes was simply mutated. The research does not resolve the question, but it builds the basis for additional research.
The science of the results and the conclusion look good. There are many instances of improper use of English that were too numerous for me to point out.

Author Response
Dear Reviewer:
Thank you very much for your letter and for the reviewers’ comments concerning our manuscript entitled “Characterization of a common S haplotype BnS-6 in the self-incompatibility of Brassica napus” (ID: plants-1357353). Those comments are all valuable and very helpful for revising and improving our paper, as well as the important guiding significance to our researches. We have studied comments carefully and have made correction which we hope meet with approval. We are very sorry for our incorrect writing in the first version. The manuscript was carefully revised and point-by-point response was provided as attachment.
Thank you and best regards.

Reviewer 2 Report
It is clear that a lot of work has gone into producing the paper by Liu et al. It seems to be mostly high quality work, which I believe to be of publishable quality. That said, I have some concerns (see specific comments below). One particular concern, only touched upon below, is the quality of the language used. I know that English is likely not the authors native one, and in general the standard of writing is good, but some of the sentences are confusing, which has an impact on understanding. In some cases I will have been explicit in the comments, but I believe it would be important to get a native speaker (or equivalent), who understands the science to correct the English in some places.
Line 16: "...also termed S haplotype." - This isn't quite so. You're describing the S-locus. An S-haplotype is a variant of the S-locus, but the terms are not interchangable.
Lines 17-19: For a 'respectively' statement to work with three species, you need three values. You only provide two.
Line 16: What do you mean by 'S -locus exist in all Brassica napus..."? Is it a recognisable region, or that it is characterized by the presence of certain genes whether functional or not? The two halves of this sentence don't work together.
Line 19: Some grammatical errors here.
Line 20: "S haplotypes" should be "S haplotype".
Line 26: "Charged"? Do you mean changed?
Line 35: Yes, single in some, but worth noting exceptions perhaps.
Line 36: I don't think 'S-locus' and 'S haplotype' are interchangable as you suggest here. Perhaps clarify this statement.
Lines 44-45: This sentence is confusing. Better to simply say "There are approximately 100 known S haplotypes in B. rapa and 50 ...". At this point, it isn't clear whether you mean identified or estimated though. Try to be specific.
Line 46: Sequence similarity may play a part in class delimitation, but dominance hierarchies need to be determined manually.
Lines 51-53: This isn't assumed knowledge. Needs an appropriate reference.
Lines 53-55: If dominance class is delimited by simply using sequence similarity, why are all of the alleles not classified thus?
Lines 62-63: What is the support for this theory?
Lines 63-64: These two sentences don't currently make sense.
Paragraph including lines 66-83: I found this paragraph difficult to follow. Many of the sentences appeared to have words missing. More explanation is needed for some concepts.
Line 86: Why is the first figure you mention a supplementary figure? Surely it makes sense to have this in the paper?
Line 86: 'Sequence collinear comparison' - I don't find mention of this in the methods. What is it, and how was it done? Further, given the suggestion that dominance classes can be defined by sequence similarity, can PCR products assessed on gel really be a suitable way of determining SCR &/or SRK? This seems dubious to me.
Line 102: Or BnS-6 like.
Line 103: How can you tell from a PCR product that the amplified fragment was from subgenome A?
Line 109: "17.68 -21.06" What do these two numbers mean? Is it supposed to be a range? What are the thresholds you use to determin if SCI falls into SI/SC/mixed mating?
Line 118: What does "S-locus widely existed in B. napus genome..." mean?
Line 119: Suggest 'natural' not nature. There are a number of such issues throughout the text. I will not point them all out, but a correction service may be worthwhile.
Line 124: So how do you go about understanding what S haplotypes are in these 35 cultivars?
Table 2: How do you distinguish whether subgenome A & C share the same SRK haplotype in the 35 unidentified lines? Indeed, how can you tell that SRK6-1 is in the C genome and not the A genome in this case?
Lines 134-136: These lines need restructuring. Also, based on what sequences (i.e. accession numbers); What are the primer sequences (table)?
Line 137: suggest 'difficult' and NOT '"different".
Line 137: What is the criteria to delimit an SCR allele then. 5 bp seems very little. Are the sRNA different? Do we understand why these are different alleles in general?
Section 2.2: Really unclear in general.
Figure 2: The scale changes in this figure are confusing.
Line 167: "significantly higher". What test? What value?
Line 169: Could be. But this doesn't help us understand the mechanism.
Line 172: How? Finding SCR is tough.
Line 175: "conserved" in this case meaning invariable? Also, "consistent" meaning what? That the sequence is identical?
Line 195: This is the first time you mention these promotor constructs. What are they and from where?
Line 200: How does this indicate loss of SI was a "common phenomenon"? Choice of language isn't clear, but you seem to be implying that loss happened frequently, which seems unlikely in this case.
Figure 4: Panel f should have cultivar type names not numbers on axis as in panel b.
Line 210: Did you? That didn't seem clear in the introduction.
Lines 227-229: I find this a little debatable.
Line 232: What 'male sterility system'?
Author Response
Dear Reviewer:
Thank you for your letter and for the reviewers’ comments concerning our manuscript entitled “Characterization of a common S haplotype BnS-6 in the self-incompatibility of Brassica napus” (ID: plants-1357353). Those comments are all valuable and very helpful for revising and improving our paper, as well as the important guiding significance to our researches. We have studied comments carefully and have made correction which we hope meet with approval. We are very sorry for our incorrect writing in the first version. The manuscript was carefully revised and point-by-point response was provided as attachment.
Thank you and best regards.

Reviewer 3 Report
Dear Authors,
The manuscript describes the development and assessment of SCAR markers for the determination of self-incompatibility using 523 inbred lines, and over-expressed the BnSCR-6 to investigate the function of BnS-6 in response to self-incompatibility. However, I found the manuscript very difficult to follow with many grammatical errors and could not understand the novelty and significance of the work. From what I could understand, the novelty of this manuscript lies in the development of SCAR markers that may be applicable for determining self-incompatibility. However, I could not find the details of the 523 inbred lines used in this study, which is necessary to interpret the data and discuss the generality of the markers. The over-expression of BnSCR-6 using the endogenous promoter reveals the function of the isolated promoters but cannot demonstrate the function of the protein. The finding itself is interesting, but I feel it requires further investigation (e.g. does it require additional cis-elements for its original function? is it expressed in the tapetum?). Also, a description of biological replicates used for some experiments is missing. I feel the manuscript can benefit by revising the manuscript to clarify the key findings and distinguish them clearly from previous researches (e.g. Zhai et al, BMC Genomics 15, 1037, 2014).
Some points of revision are as below:
Figure 1.: A negative and positive control for each primer set should be included (using lines that have been published previously), and at least a no template control. A faint band in Westar x Bing409 using SRK1300-1, in Westar using SRK7-1 can be seen.
Table 1.: The table may be good for the main text, but the PCR gel results should be included in the Supplementary files.
Figure 2.: Authors generated 20 transgenic lines but seems like only one line was used for qRT-PCR (W-3). Additional lines should be added to demonstrate the expression level variation between lines. Also, the colour code of Stem, leaf, and the different stages of Stigma cannot be distinguished. I understand that they are in different graphs, but would be easier for the readers if they were in different colours or separate figures etc.
Author Response

(The authors gave the same response as above.)

Round 2
Reviewer 3 Report
Dear authors,
Thank you for revising the manuscript. I feel the manuscript has improved significantly by correcting the writing, and can now understand some of the results that were not clear before. It may be a matter of preference, but I found the response comments easier to follow than the main text. Perhaps, it may be worth considering of rephrasing some of the manuscript text to include these.